*Research Directions:*
*One Health*

**www.cambridge.org/one**

# Existing strategies to address the risk of mosquito-transmitted dengue in the continental USA: opportunities to adopt a One Health approach

## Impact Paper

Dengue; one health; vector-borne diseases; mosquitoes; mosquito control strategies

**Corresponding author:**
Victoria A. Osasah; Email: vosasah1@jh.edu

Victoria A. Osasah[1,2] ⓘ, Eri Togami[1], Mehdi Aloosh[3], Monica Schoch-Spana[1,2] and Meghan F. Davis[1,4]

[1]Department of Environmental Health & Engineering, Johns Hopkins Bloomberg School of Public Health, 615 N Wolfe St, Baltimore, MD, 21205, USA; [2]Johns Hopkins Center for Health Security, 700 East Pratt Street Suite 900, Baltimore, MD, 21202, USA; [3]Department of Health Research Methods, Evidence, and Impact, McMaster University, Suite 2006 100 Main St. West, Hamilton, ON, L8P 1H6, Canada and [4]Department of Molecular & Comparative Pathobiology & Division of Infectious Diseases, Johns Hopkins School of Medicine, Baltimore, MD, 21205, USA

## Abstract

Recent increases in dengue cases across the region of the Americas have underscored the need for an integrated and collaborative One Health approach to address the potential of widespread autochthonous dengue in the continental USA. Improvements in the public health, social and health sectors are paramount in ensuring that communities are better protected. Furthermore, communities would benefit from effective adaptive strategies in the event of autochthonous dengue outbreaks. There is an opportunity to address existing challenges in the control of mosquitoes, public health infrastructure and funding that are necessary to recover from threats from climate-sensitive pathogens. Each component will improve preparedness toward widespread autochthonous dengue. This review provides an outline of adaptive and mitigating strategies and offers opportunities to address challenges through a One Health lens.

## Introduction

Dengue, which is caused by an arbovirus, has become the fastest-growing mosquito-borne disease in the world (WHO, 2010). In the past two decades, globally reported dengue cases have increased significantly (from >500,000 cases in 2000 to 5.2 million in 2019) (WHO, 2023), with almost 80% of the reported cases (4.1 million) occurring in the region of the Americas in 2023 (WHO, 2023). In 2024, more than 12 million dengue cases were reported in the Caribbean, North, Central and South America, exceeding the region's annual average number of 4.6 million cases (CDC, 2025c). Across the USA and its territories, the number of reported travel-associated dengue cases, which pose a potential risk of autochthonous transmission, increased by 187% between 2010 (642 cases) and 2023 (1,848 cases) (CDC, 2024e). Beyond high medical costs and high hospitalization rates, dengue produces a cascade of adverse social and economic effects, including overburdened healthcare systems, school and work absenteeism, interrupted household income, lost productivity, increased need for caregivers and poor mental health (Junior *et al.* 2022; Marczell *et al.* 2024).

While the risk of widespread autochthonous dengue transmission across the continental USA remains low (CDC, 2025c), increases in autochthonous cases—including outbreak-related cases – have been reported in Florida (532 cases: 2010–2024) (CDC, 2024e) and Texas (42 cases: 2010–2024) (CDC, 2024e). The increase also may be attributed to improved surveillance and changes in the case definitions, to address the possibility of cross-reactivity from other closely related flaviviruses, such as Zika virus, to which dengue belongs. However, these case counts are likely under-reported and may not account for under-ascertainment of cases because they are reported through passive surveillance systems (Contagion, 2025). Both states have an established presence of female *Aedes aegypti* and *Aedes albopictus* (Parker *et al.* 2019; CDC, 2024d), which can transmit the dengue virus to humans. In 2023, the Florida Department of Health reported 186 autochthonous dengue cases (Florida Department of Health, 2025), which was an almost threefold increase compared to the cases in 2022 (68 cases) (Florida Department of Health, 2025). This was the highest case count reported in more than a decade. In 2024, 85 autochthonous dengue cases were reported in Florida, the second-highest case count since 2010 (Florida Department of Health, 2025).

The public health importance of dengue in the continental USA, which has not had widespread autochthonous transmission unlike USA territories such as Puerto Rico and American Samoa (CDC, 2025e), is underscored by its case fatality of 10%–20% among untreated

people (Schaefer *et al.* 2024), and the high proportion of asymptomatic people who could potentially transmit the virus to local mosquitoes (Duong *et al.* 2015). The currently approved dengue vaccine, which should act as a primary preventive method, is age-restricted (ages 9 to 16 years old) and available for previously infected individuals (CDC, 2025d). Furthermore, there is no specific anti-viral treatment for dengue (Dengue, 2009). Approximately 25% of infected individuals will experience symptoms, and 5% will develop severe signs and symptoms (CDC, 2025b) that could range from persistent vomiting to dengue shock syndrome (Htun *et al.* 2021).

Historically, public health efforts have focused on risk-mitigating strategies at the vector ecology, environment and human host levels, but not in an integrated manner (Dusfour and Chaney, 2022). Existing strategies have shown limited success in effectively reducing the spread of *Aedes* mosquitoes due to multiple challenges, which include diminished mosquito control capacity following public health budget cuts, siloed work structures between and within jurisdictional health departments, and non-integrated work between the public health and non-public health sectors, including community residents (Hadler *et al.* 2015; Bevc *et al.* 2015; Dusfour and Chaney, 2022; Dye-Braumuller *et al.* 2022b). These gaps contribute to lapses in the effective implementation of mosquito control and could impact the risk of transmission of dengue to humans.

Current and projected modelling estimates indicate a growing population and current expansion of the *Aedes* mosquito across parts of the USA, including California, Georgia, Florida, Louisiana, Maryland, Texas (CDC, 2024d) and a projected expansion across most of the USA by 2100 (Khan *et al.* 2020). Contributing factors include climatic changes, increases in global travel and trade and growing harmful land-use practices (e.g. urbanization) (WHO, 2023). Therefore, bold and novel strategies are required to better prepare for an increased incidence of autochthonous dengue cases in the continental USA. Future strategies would need to address the multi-factorial drivers of mosquito proliferation and dengue transmission, such as vector competence, and increased temperatures and precipitation from climatic changes. A collaborative One Health approach is required to address all components of human, animal (vector population) and environmental threats related to autochthonous dengue. Incorporating measures to ensure the protection of the social and health systems in the decision-making steps of the collaborative effort can minimize harm to human health, including social well-being and mental health.

Strengthening the public health system across jurisdictional levels would be pivotal in the protection of social and health systems; priority investments could include securing sustained funding for arboviral surveillance and control, modernizing public health surveillance systems with a focus on a coordinated and multi-collaborative approach that involves unified information sharing with inter-jurisdictional, intra-jurisdictional and private partnerships during monitoring and response activities, and early, proactive, sustained engagement with communities before, during and after outbreak responses (Anggraini Ningrum *et al.* 2024). One Health collaborative efforts toward reducing the transmission of emerging and existing infectious diseases have contributed to public health success, such as the reduction in rabies cases in South Asia and Hendra virus cases in Australia (Horefti, 2023). The success can be attributed to well-coordinated responses from inter-disciplinary partners. The purpose of this paper is to review existing risk mitigating and adaptive strategies linked to mosquito-borne dengue, identify gaps hindering these strategies, and propose

opportunities to strengthen One Health approaches in addressing the gaps in preparedness toward widespread autochthonous dengue in the continental USA.

## Unified monitoring and surveillance efforts driven by multi-collaborations and access to diverse real-time or near real-time data

The interconnectedness of mosquitoes, humans and the environment in the transmission of dengue requires effective surveillance and control in all three domains to prevent or contain the threat of transmission. Beginning with mosquitoes, mosquito control requires the implementation of an integrated mosquito management (IMM), which is the recommended practice for effective mosquito surveillance and control. The challenge, however, is in the uniform implementation of mosquito surveillance and control measures across the continental USA, arising from differences in state and local laws governing the legal authority and responsibility for mosquito control, the ambiguity in the legal terms related to the responsibility of mosquito control, and the variations in the funding mechanisms for mosquito control.

In the continental USA, vestiges of the English rule in America and subsequent changes to the Constitution allowed states to maintain authority over public health threats, which was upheld by the Supreme Court's ruling in *Jacobson v. Massachusetts* in 1905 (Pepin and Penn, 2017). Additionally, the courts upheld the legal authority of mosquito control to the states in *Paris v. City of Philadelphia* (1916) (Pepin and Penn, 2017). However, the responsibility for mosquito control varies within states, counties and municipalities.

Legal responsibilities are informed by legal options, including statutory provisions and regulations, which allow state agencies to approve a mosquito control program or the establishment of a mosquito control district (Association of State and Territorial Health Officials, 2018). In addition, the provisions allow, across most states, state health agencies to require property owners to abate nuisance mosquitoes. Within each state, however, the legal responsibility for mosquito control could be centralized, which would be solely managed at the state, territorial, or district level, or it could be decentralized, managed solely at the local level. A third option is a hybrid structure, in which the state, territorial, district and local agencies bear the legal responsibility for mosquito control.

With these differing structures, variations in the express legal authority for mosquito control arise. In a statutory review by the Association for States and Territorial Health Organizations, 35 of 50 states had express legal authority for mosquito control, which ranged in their description of duties from broad to specific (Association of State and Territorial Health Officials, 2018). However, the process through which these statutory provisions are implemented was unclear. While each mosquito control structure has its strengths, some drawbacks could ultimately impact the effective and uniform implementation of the IMM. One of the benefits of a decentralized mosquito control structure is more independence for states to adapt mosquito control strategies to the local context. Prior efforts have shown success in vector control management (Schoch-Spana *et al.* 2020). However, a drawback could be the non-uniform implementation of mosquito control practices across mosquito control programs within the state (IntechOpen, 2025). While centralized mosquito control activities could aid in ensuring streamlined coordination in mosquito activities, local differences in mosquito species diversity, spread,

and growth may require the implementation of abatement strategies that are specific to the local context (Aryaprema et al. 2023).

Another challenge is the disparate funding sources of mosquito control, which could be sourced from different mechanisms. The most common is at the local level, which could come from general funds, levies, utility bills, or property taxes (common among almost half of the states) (Association of State and Territorial Health Officials, 2018). With the latter funding source, the decision-maker involved in imposing the property tax could range from special districts to voters. In some cases, these variations have impacted the establishment of mosquito control districts, necessary for mosquito surveillance, investigation and abatement (Dickinson et al. 2022). These forms of funding are often not sufficient and require additional funding from the state or federal budget (Association of State and Territorial Health Officials, 2018). An additional consequence of these disparate funding structures is the shortage of vector-borne disease specialists, including entomologists who could aid in identifying mosquito species (Almeida, 2017). A dearth of entomologists in New York City, including during the first outbreak of West Nile Virus in 1999 (Miller, 2001) contributed to delays in detection and response efforts. The deficiency is usually largely due to budget cuts that have stymied the enhancement of training and degree certifying programs for entomologists, including research required to enhance mosquito surveillance and control (National Association of City and County Health Officials, 2021).

The different structures for the responsibility of mosquito control and the diverse funding mechanisms create challenges in implementing IMM (CDC, 2024c). For instance, in Texas, which operates under a decentralized system in which the local government and property owners share the responsibilities of mosquito control, private property owners are required to abate nuisance mosquitoes. However, local health authorities do not have jurisdiction to enter and conduct mosquito abatement in inhabited private properties (Association of State and Territorial Health Officials, 2018). In addition, in Texas, mosquito control districts can only be created following a formal request and a vote (Dickinson et al. 2022). The creation of a mosquito control district could lead to an increase in taxes in that county, which may lead to a reluctance to establish it (Dickinson et al. 2022).

Furthermore, statewide mosquito control practices could differ across inhabited private properties based on the property owner's knowledge of effective practices. Inconsistencies in the practice of mosquito control and the use of ineffective measures could hinder the progress of previous efforts by groups such as private residents, pest control companies, scientists and government authorities, who often work separately to abate the risk from mosquitoes.

## Opportunities

A key problem with the existing practice of mosquito control is tied fundamentally to the structure of responsibility related to mosquito control. The opportunities for improvement outlined in this section pertain to a scenario in which a decentralized structure of mosquito control exists, based on the presence and distribution of *Aedes aegypti* or *Aedes albopictus* in the geographic region, including human mobility patterns, and the region's prominent economies (e.g., economic activities that could increase the number of dengue carrying mosquitoes such as travel and tourism). In this scenario, the primary goal is to ensure a unified system that engages stakeholders from across disciplines and sectors.

Using a targeted One Health approach, a few strategies could improve upon gaps in preventing widespread dengue transmission to humans. The first strategy would include the implementation of a unified platform for uniformly utilizing surveillance data from diverse sources, including geoclimatic, sociocultural, human mobility, economic and post-disaster-related data based on recent environmental changes, (e.g., disasters such as wildfires that could create a more favorable breeding environment for *Aedes* mosquitoes) (Webb et al. 2021). Other relevant data should include vector competence, region-specific species distribution and genomic data, integrating information on historical and proposed effective vector control methods. The second strategy would involve integrating and sharing the information across a team of experts comprising transdisciplinary stakeholders, who can utilize the information to create a synthesized and region-specific approach to reduce *Aedes aegypti* or *Aedes albopictus* in their communities.

Using real-time or near-real-time data could lead to effective collaborative decisions that produce timely cascading effects on reducing the transmission of dengue. Actor groups, including entomologists, clinicians, ecologists, epidemiologists, social scientists, the private pest control industry and community members, could strategically develop integrated interventions informed through these collaborative efforts. For instance, using evidence-informed approaches between community members and scientists could lead to successful interventions, as observed in the Citizen Action Through Science model (AcTs) (Johnson et al. 2018). Such integrated efforts of community members in Maryland and scientific advisors from New Jersey contributed to reducing *Ae. albopictus* by using cost-effective interventions such as oviposition traps and canola oil. However, the use of canola oil was not without harm to other aquatic life that may have been drowned. Additional efforts could include the use of targeted, non-toxic biopesticides such as *Bacillus thuringiensis*.

Another successful collaboration involved a partnership between academia and community residents. The project was conducted in East Tennessee, in communities where there was limited *Aedes* mosquito surveillance for the vector of La Crosse virus, and limited community engagement. By providing training in *Aedes* mosquito surveillance and abatement to middle and high school educators, who subsequently trained their students, researchers observed comparable, successful surveillance techniques and a significant decline in the number of *Aedes* larvae. This achievement occurred by removing neighborhood litter, including container-like breeding sites for larvae, as well as using oviposition traps with bovine liver powder and water (Day and Trout Fryxell, 2022). Such non-toxic, target-specific collaborative efforts can ensure that communities experience minimal harm from mosquito control measures (Stepan, 2022).

An additional example could be the use of region-specific, culturally appropriate, multi-collaborative solutions, including local One Health experts in reducing the risk of mosquito-borne dengue. The benefit of a local entomologist includes an expert who is well-versed in the diversity of local species, leading to timely detection and response of disease-causing mosquitoes. In collaboration with other local One Health experts, more localized solutions that are environmentally safe and targeted to a specific mosquito species could be employed. In addition, pest control companies could work alongside academia and local community members to ensure sustainability efforts in reducing mosquito-borne dengue.

These initiatives would benefit from considering potential challenges in engaging communities that have historically been

economically, socially, or politically marginalized, including communities where fear related to immigration status could impact access to clinicians or local public health officials (Martinez *et al.* 2015). An emphasis on early and direct engagement with local community members who are key resources to their communities could bridge this gap. Such a success was described in a study conducted within a predominantly Latine/Latinx community in South Texas comprising unincorporated homes, also known as *colonia,* on the border of Texas-Mexico. These communities were at a high risk of mosquito transmission due to poor infrastructure, such as unpaved roads that could pool standing water following rain showers (Juarez *et al.* 2022). These communities that were previously labeled hard-to-reach substantially contributed to planned interventions. Working with trusted collaborators in the communities to co-develop sustainable solutions could support effective and sustained engagement with community members. However, this would require long-term engagement, which should include cultural humility, with the community. The benefit could include a trusting and mutually rewarding relationship that advances success in public health preparedness within and around these communities.

### A consistent and robust public health funding system to prepare for future dengue-related public health emergencies using a One Health approach

Historically, the public health field has experienced several decades of gutted budgets that often strip the sector of core public health programs, which are integral in the protection of the health of communities. This has led to the de-prioritization of key public health programs, such as disease surveillance, detection and control (NACHO, 2025). In the continental USA, the gutting of public health budgets and the infusion of funds to the sector have followed a similar cycle that revolves around the infusion of more funds, typically during public health emergencies such as the COVID-19 pandemic or the Zika epidemic. Once an epidemic or pandemic is contained, then memories of the crisis fade, and the public health budget faces cuts. With tremendous cuts to the public health budget, the median *per capita* for public health expenditure at the local level as of fiscal year 2015 was $33.50, with variations across states. The median *per capita* in 2015 was similar to the fiscal year expenditure in 2008 ($33.71 *per* person), accounting for inflation (TFAH, 2025). This lack of growth in funding, and potential for substantive cuts in 2025, has occurred in the face of increasing public health threats, such as climate-sensitive pathogens like dengue and poor public health infrastructure to better prepare for such threats. Existing evidence has shown that inconsistent funding can contribute to large outbreaks and epidemics, responses that are reactive instead of proactive, and a poorly prepared public health sector (Alfonso *et al.* 2021). Furthermore, sustained cuts harm funding necessary for scientific investigations to study the impact of public health interventions (National Academies Press, 2012).

Funding for arboviruses, such as dengue, has equally faced significant budget cuts. For instance, between 2004 and 2012, federal funding was reduced by 61% for arbovirus-related programs such as surveillance of animals (e.g., birds infected with West Nile Virus [WNV]), mosquitoes and arboviral human diseases (IntechOpen, 2025). A pre-post assessment of surveillance and laboratory capacity by the Council for State and Territorial Epidemiologists, showed a decreased capacity for mosquito surveillance and laboratory testing of arboviruses such as dengue

across local health departments (LHDs), de-prioritization of human disease surveillance for WNV or dengue, and an absence of a coordinated disease surveillance system for other arboviruses, including dengue (Hadler *et al.* 2015; IntechOpen, 2025). The LHDs with reduced support through funding cuts also had limited resources to conduct mapping of the distribution and capacity of *Aedes aegypti* or *Aedes albopictus* to transmit disease, which is integral for IMM. These cuts go beyond the elimination or reduction of mosquito programs and into staffing adjustments, which exacerbate the situation of insufficient staffing of experts, such as entomologists, who are essential in conducting mosquito surveillance and control activities (IntechOpen, 2025).

In 2011, a proposal to cut the budget for vector-borne disease programs, including mosquito programs, was strongly opposed by the scientific community and other advocates for mosquito control (Vazquez-Prokopec *et al.* 2010). If the budget cut had been approved, significant setbacks to mosquito programs would have occurred, further periling mosquito surveillance and control. Considering that dengue is a climate-sensitive pathogen, budget cuts or the elimination of funding related to climate-adaptive strategies could heavily impact the risk of exposure to *Aedes* mosquitoes that could carry dengue in communities residing in climate-vulnerable regions and communities with low socio-economic status (de Jesús Crespo *et al.* 2021), following natural disasters (Acosta-España *et al.* 2024; Moise *et al.* 2024). Historically, case examples of cutting such mosquito programs have shown the impact on communities, such as the example of an LHD that reduced its mosquito control programs and subsequently faced the presence of the virus causing Eastern Equine Encephalitis among local mosquitoes (CDC, 2005). The key problem with the funding for vector-borne diseases, such as dengue, is that such diseases are prone to funding cuts, based on the under-estimation of the occurrence, burden and severity of these diseases (incidence, prevalence and mortality) in the continental USA due to inaccurate assessments of their impact (LaBeaud and Aksoy, 2010).

At the local level, public health funding comes from a mix of federal, state and local sources, with approximately one-fourth from the federal government and one-fifth from the state on average pre-COVID pandemic (NACHO, 2025). A key roadblock that exists at the local level is the structure of the grant funding that could be disbursed either as a categorical or block grant. The former, which is more common, is silo-prone due to the specific objectives tied to it. A categorical grant is also the preferred type of grant for Congress to approve due to its program or disease specificity and the ability for its impact to be measured (National Academies Press, 2012). However, it is averse to cross-branch or inter-sectoral collaborations due to the rigidity in its funding objectives. The block grant, which is more flexible than the categorical grant, is prone to budget cuts due to the nature of its broad scope in its program objectives (U.S. Congress, 2003).

### *Opportunities*

A hybrid structure of the block and categorical grants could allow for flexibility in program objectives that would enhance cross-sectoral collaborations vital for a One Health response in dengue outbreaks, supported by region-specific needs-based assessment (Dye-Braumuller *et al.* 2022b). Furthermore, moving from the use of traditional economic models that focus on specific public health areas to the use of a One Health framework in the economic evaluation models could aid in determining an ideal public health budget. An application of such a model could show the cost-

benefits of funding public health through cross-sectoral collaborations across the domains of human, animal and the environment in the prevention of dengue. In addition, evidence of the cost-saving effect of the application of the One Health framework, as well as the cost-saving effect of robust and systematic disease surveillance systems for dengue, will support a proactive approach compared to the costs of outbreak response efforts (Vazquez-Prokopec *et al.* 2010). The evidence could strengthen advocacy efforts of groups such as the American Mosquito Control Association (AMCA) and the Entomological Society of America (ESA). These groups have advocated on Capitol Hill for sustained funding for mosquito surveillance, control and research (Dye-Braumuller *et al.* 2022b).

These efforts could support legislative bills that aim to increase and maintain sustained allocation of federal funding for public health, which have successfully passed in the past. Examples include the Mosquito Abatement for Safety and Health (MASH) Act in 2003 and Strengthening Mosquito Abatement for Safety and Health (SMASH) Act in 2016 (Library of Congress, 2015). Creative ways of adapting to the boom-and-bust funding cycle could strengthen an evidence base for more funding for mosquito surveillance and control. For instance, emergency funds from the Zika epidemic led to the establishment of Regional Centers for Excellence, aimed at training new entomologists, and enhancing research into innovative mosquito control measures (CDC, 2024b). In 2019, a federal law reauthorized funding for these centers, including the implementation of a national strategy for vector-borne disease control (CDC, 2025a). The framework was a starting point, but without flexible funding that allows cross-disciplinary and inter-sectoral One Health collaborations, these efforts could fall short.

Another strategy could involve the establishment of a minimum public health budget for vector-borne diseases (National Academies Press, 2012), still utilizing a One Health framework, estimated from the core public health needs and programs with considerations of cross-collaborations involving the domains of human, mosquito and environment across multiple sectors. The minimum public health budget could enhance public health response toward dengue beyond region-specific needs, with a robust range that would allow for fluctuations in federal budget cuts (National Academies Press, 2012).

Lastly, a unified advocacy effort that includes academia, mosquito industry stakeholders, such as the AMCA and the ESA, public health practitioners, policy makers and community members, led by science-driven evidence and real-life impacts from a community perspective could support increased funding for arboviral surveillance, control and research in Congress. Such an effort was successful in 2011, when impacted parties successfully advocated for reversing the proposal to cut funding for a vector program (NCBI, 2012).

## Novel technologies and region-specific interventions through a One Health perspective

Best practices in mosquito prevention and control involve the application of integrated mosquito management, which includes consistent community engagement, mosquito surveillance including trapping and mapping of mosquito habitat, the use of multiple forms of mosquito controls (e.g. physical, biological, source reduction and chemical controls), and monitoring of the efficacy and resistance of insecticides. Additional surveillance activities include testing mosquito pools for dengue serovars (Scott *et al.*

2021). Traditional methods for mosquito control typically include source control (removal of potential breeding sites for mosquitoes, including phytotelmata that can act as containers with stagnant water or stagnant water in containers), biological controls (e.g. the use of predatory fish such as *Gambusia* spp or predatory plants), larvicides (e.g. bacterial larvicides, insect growth regulators or oils and monomolecular films), and adulticides such as, pyrethroids, organophosphates (American Mosquito Control Association, 2025).

These traditional methods have varying levels of efficacy, and some carry considerable risks to aquatic life with potential concerns to human health. Furthermore, adulticides could contribute to insecticide resistance among mosquitoes (Weng *et al.* 2024). In addition, they are non-specific, and while high-quality epidemiological studies are yet to be developed to study the impact on human health, cross-sectional studies suggested potential caution on environmental exposure of pyrethroids to humans (Andersen *et al.* 2022). Among aquatic animals, multiple studies have shown a link between pyrethroid exposure and early developmental deficiencies (Brander *et al.* 2012; Jaensson *et al.* 2007). Best practices via the IMM approach include the use of adulticides only when necessary, based on findings from surveillance activities or in response to an outbreak. The use of adulticides requires specific requirements, which include target specificity, community buy-in, insecticide resistance testing and environmental compatibility (American Mosquito Control Association, 2025).

Some of the challenges with the implementation of IMM at the local level include insufficient and inconsistent funding following sustained budget cuts that lead to cutting mosquito control programs and a dearth of experts necessary to ensure that IMM is correctly conducted. For instance, in a survey of the mosquito capacity for Zika virus, a flavivirus within the family *Flaviviridae* to which dengue belongs, it was revealed that while IMM was accepted as part of best practice among LHDs, approaches to mosquito surveillance and control lacked standardization, with deficiencies in the approach to ensuring adequate mosquito sampling and trapping, and with insufficient application of effective mosquito control measures consistent with best practices. Gaps existed in an integrated and coordinated approach to cross-jurisdictional mosquito surveillance and control efforts. In addition, the standardization of uniform and effective mosquito surveillance within and across jurisdictions was lacking (IntechOpen, 2025). Some LHDs reported conducting patchwork efforts in mosquito control to make up for deficiencies arising from budget cuts. LHDs also lacked expert staff vital to conducting effective mosquito control.

Furthermore, subjectivity guided the establishment of action thresholds, which was inconsistent with the best practices for IMM. In addition, few mosquito control programs conducted mosquito testing as a reactive approach rather than a proactive approach, with some LHDs making decisions without using the findings. Limited community knowledge and community buy-in (IntechOpen, 2025), due to concerns on the use of chemicals and other methods of abatement perceived to be unsafe to humans and the environment, impeded engagement and effective vector control strategies consistent with the IMM.

Surveys of vector control programs in the Southeastern region of the continental USA, a region with states that have been disproportionately impacted by dengue, highlighted challenges especially among vector control programs that are not tied to a local health department, in conducting adequate mosquito surveillance activities. Such deficiencies in mosquito surveillance included a lack

of testing for dengue serovars in the mosquito pool (Dye-Braumuller *et al.* 2022a). In addition, less than half of the vector control programs performed insecticide resistance testing. Similar findings were observed in a national survey of 483 local vector control programs (LVCPs), in which 72% of LVCPs indicated that their mosquito surveillance and control, integral in IMM, needed improvement (National Association of City and County Officials, 2020). Furthermore, only 31% ($n = 483$ from 48 states) of LVCPs reported the application of insecticide resistance testing as part of best practices (National Association of City and County Officials, 2020).

Combined, these gaps could weaken mosquito surveillance and control. For instance, a few studies showed the influence of insecticide resistance on the vector competence of local mosquitoes in Florida, suggesting knockdown resistance mutations for genes encoding transmembrane proteins in *Aedes* mosquitoes (Chen *et al.* 2021). This mutation ultimately decreased sensitivity to pyrethroid, which correlated with increased susceptibility to a dengue strain in mosquitoes (Chen *et al.* 2021).

### *Opportunities*

A coordinated information system, among LHDs operating in a decentralized structure, which enhances the sharing of information, including applied or field methods for mosquito surveillance and control, could aid in ensuring that decision-makers leading mosquito control programs understand the gaps that could exist in their efforts toward an effective mosquito control strategy. Furthermore, a standardized protocol for mosquito surveillance, such as sampling methods and a standardized method of establishing action thresholds, should be implemented across mosquito control programs while adjusting for region-specific differences to address species diversity and distribution by region. Smaller or less funded vector control programs and LHDs that have a wide distribution of *Aedes aegypti* or *Aedes albopictus* could be prioritized in the establishment of a collaborative workgroup.

Alternative measures of mosquito control could be applied, such as genetic-biocontrol measures. These methods include Sterile Insect Technique, the use of *Wolbachia* bacteria in mosquitoes, as both a mosquito population and pathogen suppressant, and Oxitec OX5034 *Aedes* mosquitoes, using a method known as Release of insects carrying a dominant lethal gene. These methods are target-species specific, reduce the mosquito population, and have been adopted as non-traditional control methods. Non-randomized studies in Brazil and Colombia showed the success of using *Wolbachia* mosquitoes in reducing the transmission of arboviral diseases (Côrtes *et al.* 2023). However, a large population of *Wolbachia Aedes* mosquitoes are required to prevent transmission of dengue strains capable of evading *Wolbachia*, since *Wolbachia* strains show varying abilities in preventing transmission (Mustafa *et al.* 2016).

While these forms of genetic-biocontrol pose minimal environmental harm, the inherent nature of the methods could pose a threat to the ecological balance under less well-managed scenarios. Furthermore, community concerns on the impact of this alternative on mosquitoes and the ecosystem could hamper its effectiveness. For instance, following a dengue outbreak in Key West, Florida, between 2009 and 2010, surveyed community residents on the use of *Wolbachia* in mosquitoes expressed differing support as a mosquito control measure that varied based on educational level and racial and ethnic category (Ernst *et al.* 2015). A participatory One Health approach could address the

gaps in efforts to control mosquitoes by advancing solutions that have strong community buy-in. Other newer technologies, such as Next-generation CRISPR population suppression methods and anti-pathogen effectors in the *Aedes* mosquitoes, could be considered, ensuring One Health approaches that include considerations for humans, animals and the environment.

Building upon transdisciplinary collaborations, experts such as synthetic biologists, epidemiologists, environmental health specialists, laboratorians, entomologists, ecologists, community members, economic analysts and private pest control can develop an effective strategy that utilizes novel technologies to control mosquitoes. Developments from synthetic biology can be used to identify interventions that cause minimal harm to mosquitoes. For instance, *Aedes aegypti* mosquitoes have been genetically engineered to reduce their viral load of a dengue serovar (DENV-2), ultimately creating a resistance to dengue transmission (Weng *et al.* 2024). Since the ecology and genetics of mosquitoes vary across regions and within cities (Bradt *et al.* Bradt et al., 2019), region-specific and city-specific applications of genetic engineering that are not deleterious to mosquitoes, humans and the environment could help control mosquitoes through collaborations with local experts.

### Strengthening and modernizing human disease surveillance for dengue, using diverse data sources and an adoption-wide approach in using novel technologies that support a One Health response

Previous epidemics and pandemics highlighted a need for timely data from traditional and non-traditional sources. There are a few challenges to achieving a streamlined approach to data acquisition and utilization. One challenge is the reliance on passive surveillance systems for both traditional and non-traditional data sources, which depend heavily on reporting disease events by health providers and laboratories to health departments. For instance, this form of reporting of human cases by health providers could be prone to under-reporting, including delays in timely reporting, misdiagnosis and underdiagnosis., Since people can acquire asymptomatic dengue, people who do not seek medical care will be missed. Furthermore individuals with minimal resources, including people with limited financial resources, limited means of transportation to healthcare providers, and a lack of legal immigration status, to seek medical care, and with higher dengue susceptibility based on occupational, environmental or immunological exposures, may not be identified by the healthcare system (Soto *et al.* 2023) until they reach a severe stage of the disease, which could be fatal.

Another key challenge is the dismal investment in public health infrastructure that requires the modernization of existing disease surveillance systems, which would allow for more timely monitoring and response efforts. Disease surveillance systems are often not designed to be efficiently interoperable with other external data systems, and may be poorly designed for flexible use of the data fields during disease outbreaks (Lau *et al.* 2021). A step in the right direction was the five-year federal grant, Public Health Infrastructure Grant (PHIG), which was aimed at strengthening public health infrastructure. However, the grant funding prematurely ended in March 2025 (CDC, 2024a). The consistent gutting of public health funding has contributed to a lag in the modernization of disease surveillance systems necessary for a uniform collection and storage of data from diverse sources for emerging climate-sensitive pathogens like dengue (Kadakia and Desalvo, 2023). Furthermore, the current structure of funding that has contributed to intra-jurisdictional and inter-jurisdictional siloed work is not

conducive to a streamlined and integrated system of data sharing for a One Health response.

## Opportunities

Existing infrastructure, such as the disease surveillance systems, can benefit from being on interoperable and flexible platforms, which would allow for a streamlined use of diverse, novel, traditional and non-traditional data sources (e.g., mobility or spatio-temporal data). This enhancement would also allow for inter-disciplinary and transdisciplinary monitoring and response to dengue, in addition to the use of machine learning processes such as Artificial Intelligence (AI). To bridge the gap of siloed collaborations, existing One Health Workgroups and consortia such as the North America One Health University Network or a collaborative group similar to the National COVID Cohort Collaborative (NC3), which has been successful in overcoming barriers to data sharing and minimizing siloed work, could be leveraged to enhance preparedness toward widespread autochthonous dengue (Guralnik, 2024; Ohio State University, 2024). Through collaborations with experts, including epidemiologists, entomologists, climate scientists, laboratorians, microbiologists, mobility data specialists and computer technologists, a more robust response to dengue could be produced.

For example, using integrated information from multiple data sources, such as spatio-temporal patterns, human mobility data, and local economic data, machine learning processes could better refine insights into future predictions of the incidence of dengue cases and potential outbreak locations (Anggraini Ningrum et al. 2024). Findings from simulated or real post-effects of climate-related events, such as flooding, using AI could inform public health officials of potential areas where people could be most at risk of exposure to Aedes mosquitoes or where potential dengue outbreaks could occur. LHDs could complement gaps in technical expertise, such as in AI, by extending partnerships with academic and other research institutions. For example, during the COVID-19 pandemic, the use of AI in predicting epidemics showed high success indicating its utility in epidemics (Wang et al. 2021). The use of non-traditional surveillance data, specifically wastewater, was also vital in estimating COVID-19 and mpox cases. This was made possible by transdisciplinary collaborations beyond the public health sector. Prior successes can be replicated with dengue. Lastly, including active disease surveillance along with passive surveillance could aid in identifying potential changes in the epidemiology of the disease in non-endemic regions of the continental USA. However, changes to existing public health infrastructure, requiring sustained and increased resources and monetary investment, are necessary. If successful, the modernization can contribute to a robust surveillance system better equipped for dengue.

## Enhancing responses to public health emergencies to protect and promote the resilience of social and health systems

The COVID-19 pandemic and the mpox outbreak in 2022 have underscored the far-reaching effects of emerging infectious diseases on communities, which include financial and economic loss, periods of unemployment due to illness, and the adverse effects from poor social well-being and compromised mental health (Almeida, 2017). The most affected populations included individuals at the intersection of social and economic identities, such as people identifying as immigrants and people of color, and people who

already faced adverse social and health outcomes. Poor communication and limited public engagement played a pivotal role in outbreak response efforts and may have contributed to misinformation across social systems (United States Government Accountability Office, 2024; Kim and Kreps, 2020). Misinformation during the COVID-19 pandemic impacted the community buy-in of public health messaging, essential to stop the spread of the virus. Vulnerable components that were harnessed for misinformation involved the social and health systems. Historical and existing patterns of distrust of public health authorities and health systems were further exacerbated with an uptick in misinformation (United States Government Accountability Office, 2024; Jennings et al. 2021).

## Opportunities

A key approach to avoid similar problems with dengue involves an engagement in a direct and consistent relationship between the public health system and the public in the preparedness efforts toward dengue. This will also require more data transparency, such as the sharing of more timely, actionable public health information to the public, especially at the local level. The proverbial seat at the table should be adopted, uniting public health officials and actor groups from communities, especially communities that have experienced historical harm by the public health system, in the key planning and decision stages of public health interventions. The groups can include agricultural immigrant workers who, due to structural inequities, immigration status and political determinants of health, suffer disproportionate impacts from diseases like dengue (Msellemu et al. 2024). In addition, the group could include individuals or representatives of people living in poor housing conditions, since they experience a higher risk of mosquito-borne disease transmission (Chastonay and Chastonay, 2022; WHO, 2017).

The approach should also include the identification of information sources through which health information is predominantly shared, and in collaboration with these media platforms, prior to public health emergencies. This could include collaborations between the public health system, epidemiologists, community members and custodians of social media platforms to advocate for a legislative bill that will ensure a unified approach to the dissemination of trusted and scientifically rigorous information via social media platforms. Such a legislative bill could include safeguarding evidence-based information using a series of standardized metrics consistent with ensuring the accuracy of the findings of scienctific information. These metrics could indicate if the shared information meets the criteria for rigorous and evidence-based information. An example would involve shared information that is supported by a well-designed, peer-reviewed systematic review and meta-analysis. This approach ensures that misinformation through social systems is minimized.

Another key policy-related tool that could enhance One Health collaborations toward addressing misinformation would involve effective data governance regulations, which would restrict the potential dissemination of generative AI-health information from non-trusted sources that contribute to misinformation. The benefits of combating misinformation on different fronts could include stronger and more effective public health interventions that could better protect the health systems. An example of a similarly successful operation is the European Code of Practice on Disinformation that was approved by the European Commission and the European Board for Digital Services (European

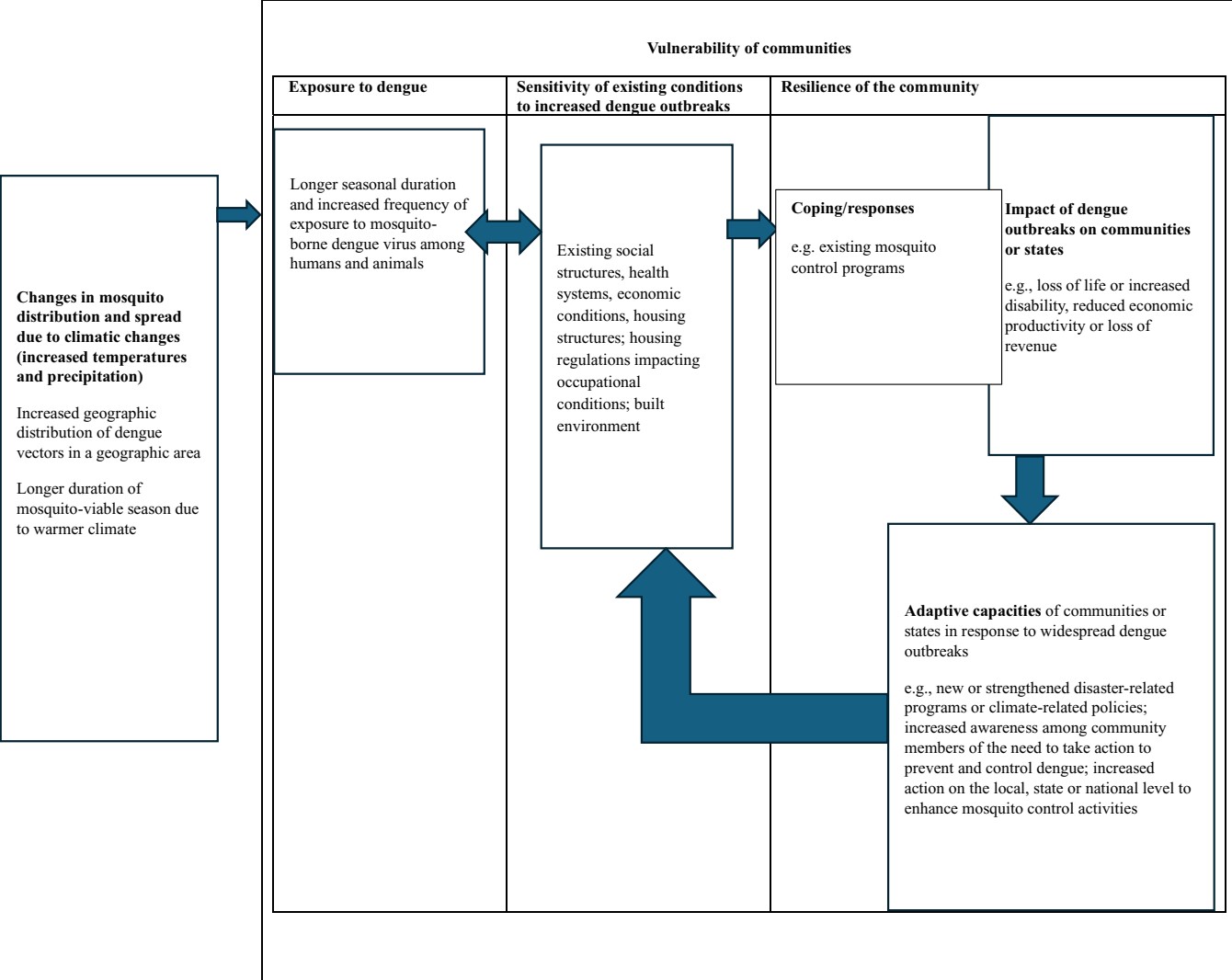

**Figure 1.** Factors linked to a community's vulnerability to dengue.

Commission, 2025). It involved a multi-collaborative effort that included custodians of social media platforms, civil society organizations, the advertising industry and fact-checking organizations. Following support for the Code, it will serve as a benchmark for assessing compliance of online information to minimize disinformation (European Commission, 2025).

Engaging early with affected populations in designing public health interventions, with considerations of their diverse socio-political circumstances, can improve health and social outcomes from dengue. It can also minimize misinformation (Sundelson *et al.* 2023). Community engagement, for instance, if performed through the lens of cultural humility and participatory approaches, can enrich outbreak control measures toward dengue. It can also strengthen health education campaigns by crafting meaningful messages tailored to specific audiences and foster community-led advocacy to strengthen support for the public health infrastructure. An example of such success was observed in the Citizen Action Through Science (AcTs) model, where the population of *Ae. albopictus* was reduced significantly through collaborations

with community members and the scientific community (Johnson *et al.* 2018). Other examples include a community-academia collaboration in East Tennessee to reduce the transmission of La Crosse virus in the community by the application of non-toxic, environmentally safe interventions for mosquito abatement. Another example includes the Florida *Aedes* Genome Group, which engages college-level students in mosquito surveillance and data collection. In both instances, the groups became more engaged and knowledgeable about the standard process of mosquito surveillance and control, including their role as residents in mosquito control (Wagner-Coello *et al.* 2024).

Ensuring the continued resilience of the healthcare system during a potential dengue outbreak or epidemic would involve a multi-sectoral collaboration among the public health system, healthcare system, community members, custodians of social media platforms, computer technologists, environmental health specialists, marketing and/or advertising industry members, mobility data specialists, entomologists and political data strategists. This collaboration could help in minimizing

misinformation and in predicting patterns that could increase the risk of misinformation. This multi-sectoral collaboration using a One Health lens could identify groups that are most at risk, ultimately informing strategies toward effective outreach efforts via multiple media outlets. Simulated scenarios of public health systems and healthcare systems weathering a potential drawback or a surge of misinformation hampering monitoring and response efforts to dengue due to shifts in the political climate can be beneficial as a One Health related strategy among experts from the healthcare system, public health system, computer technology field and political analysis field.

The engagement of these communities will require the use of diverse players and participatory tools, recognizing that these communities can comprise people with diverse lived experiences and with a range of socioeconomic or sociopolitical status: people who experience homelessness, disaster-related internally displaced people, people living in climate-vulnerable communities, people with historical harm caused by the public health system, and people without legal immigration status (Cheung *et al.* 2020). This can involve the use of social media and direct-to-indirect outreach through linkage partners such as *promotores de salud* (Watkins *et al.* 2023). One Health collaborations at the jurisdictional levels have shown the effectiveness of utilizing a unified approach in preventing adverse disease outcomes. Such examples include One Health collaborations at LHDs in New York toward efforts to address antimicrobial resistance. The strength of these collaborations lies in the diversity and inclusivity of the ideas, including the high potential for community buy-in resulting from early engagement (NACCHO, 2024).

### Gaps in current adaptive strategies toward climatic changes favorable to mosquito growth

In public health, vulnerability is often measured through the lens of 1) a community's exposure to a climatic condition, for example, rising temperatures favorable to climate-sensitive pathogens such as dengue 2) a community's sensitivity to the climatic condition and 3) a community's adaptive capacity to the change. Figure 1 illustrates an example of factors linked to vulnerability for dengue (Turner *et al.* 2003). Communities most vulnerable to adverse climatic conditions would typically have more exposure, be more sensitive and experience a more adverse impact than other communities. Furthermore, they would have a limited capacity to adjust to the climatic conditions. These communities tend to be predominantly inhabited by individuals identifying as people of color, individuals who hold a low socioeconomic status, and individuals who are immigrants (Gamble *et al.* 2016).

Adaptive strategies that address the impact of climatic changes allow communities to adjust to adverse changes from the climate, including the impact of climate on vector-borne diseases like dengue. In mainland USA, 28 of 48 states and the District of Columbia implemented state-wide climate adaptation plans (Georgetown Climate Center, 2024). Plans included solutions for a rise in sea levels and flooding. However, only Alaska included direct provisions in its plan to enhance vector-borne disease surveillance as part of its adaptive strategy (Georgetown Climate Center, 2024). While some state plans included provisions to enhance certain public health activities, the historical and current budget cuts that impact public health funding could hinder efforts to address climate-sensitive pathogenic transmission. Instead, the creation of an innovative One Health funding scheme specifically to address the risk of the spread of climate-sensitive pathogens is

needed. Establishing competitive, flexible funds separate from the public health budget could ensure proper attention to the role of climatic changes in the transmission of dengue. For instance, encouraging a One Health approach to dengue preparedness would allow for strategic alignment of resources across sectors and across departmental branches, to better prepare for the emergence of climate-sensitive pathogens (NCBI, 2012).

### Conclusion

Several challenges have been identified across LHDs in the surveillance and control of mosquitoes, as well as in dengue surveillance among humans. A key challenge involves the non-uniform implementation of effective mosquito surveillance and control measures that would benefit from a standardization of practices. Creating a collaborative consortium such as NC3 or Europe's Infravec, to develop novel vector control measures for diseases such as dengue or Zika could be beneficial in addressing such challenges. This consortium could span through regional authorities, aiding in the streamlined process of coordinating a uniform and region-customizable mosquito surveillance control (Infravec2, 2025). A strengthened public health system, capable of utilizing novel technologies that aid in real-time or near-real-time predictions of dengue can better prepare the public health system for the potential of widespread transmission of autochthonous dengue, which could in turn bolster the social and health systems. A One Health approach focusing on modernizing existing disease surveillance systems, engaging with communities early in the development of interventions for a strong community buy-in, and using evidence-informed methods in mosquito surveillance and control could mitigate the risk of dengue in the continental USA. Sustained investment and creative financing structures are necessary for effective change.

**Data availability statement.** Data availability is not applicable to this article as no new data were created or analyzed in this study.

**Author contributions.** Conceptualization – VO, ET, MS, MD; Writing and reviewing – VO, ET, MA, MS, MD; Visualization – VO, ET; Supervision – MA, MS, MD.

**Financial support.** ET was supported by a grant from the U.S. Centers for Disease Control and Prevention, National Institute for Occupational Safety and Health to the Johns Hopkins Education and Research Center for Occupational Safety and Health (award number T42 OH0008428).

**Competing interests.** The authors have no conflicts of interest

**Ethics statement.** Ethical approval and consent are not relevant to this article type

### Connections references

**Fernandez de Cordoba Farini C.** (2023) How can we improve and facilitate multi-sectoral collaboration in warning and response systems for infectious diseases and natural hazards to account for their drivers, interdependencies and cascading impacts? *Research Directions: One Health* **1**, e11. https://doi.org/10.1017/one.2023.4.

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
