## [Reviewer Report]

This manuscript brings up some key issues and topics relevant to mosquito control and building a responsive and inclusive public health plan to manage mosquito-borne disease. Please see below for some specific comments, corrections, and recommendations to improve the manuscript.

Lines 33-38- Hawaii does not have widespread autochthonous dengue transmission. The last outbreak in 2015 was ~260 cases and limited to a small are of one part of one island. Prior to that the case numbers were on par with the outbreaks noted earlier in the paragraph in Florida in limited locations on one island and are much less frequent than occur in Florida. Sangwoo et al. 2022 Frontiers in Tropical Diseases reviews this information. As indicated Puerto Rico fits this description, but so do other US territories. The Sangwoo et al. 2022 article will have some information about those too, but the CDC website should as well.

Lines 83-87- While this paragraph is tempered with some “could” and “may” a sentence or two could probably be included that encapsulate that many of these nuisance mosquitoes are also potential vectors, and sometimes displaced native, typically non-human biting mosquitoes that served a similar ecological role as pollinators, prey etc. For example, when Aedes albopictus arrived in Florida it displaced many native mosquitoes, Aedes triseriatus, Wyeomyia mitchellii, Orthopodomyia signifera and many others. A great overview of this was published in 2021 by Phil Lounibos as an IFAS extension doc, ENY2057. Most of these species rarely bite humans/transmit disease and still exist in small populations in Florida, or in nearby states and could potentially refill these niches if Aedes albopictus was eradicated.

Line 104- Albopictus should be lowercase

Line 105- not sure that canola oil is necessarily “non-toxic” in the context of mosquito control. When used as a mosquito larval control, canola oil/other oils work by suffocating/drowning mosquito larvae that need to obtain oxygen at the water’s surface. This killing is not limited to mosquitoes and any aquatic organism that obtains oxygen in a similar manner will be killed and some that land on the water surface and get stuck in the oil will also drown. Bacillus thuringiensis (Bti) might be a better example as it is non-toxic to almost anything that isn’t a mosquito or fly, and is even sold for home use. The National Pesticide Information Center will have more information.

Line 108- I don’t really think that this paragraph illuminates the “Challenges with core capacities for mosquito surveillance and control” header. Some suggestions: Line 109- any takeaways or specifics on the improvements LCVPs noted. Line 110: What gaps in capacity? Line 111: What is the existing funding structure, earlier paragraphs indicate that “legal authority responsible for mosquito control varies across states”, so I am assuming funding does as well. So, with these inconsistencies does changing the existing funding structure really work as a blanket statement. Possibly so, but we don’t have enough background to evaluate.

Line 118 even with proper insecticide resistance testing you can get extensive insecticide resistance in the populations, because alternative insecticide options are limited and often expensive. So, many programs have nothing to switch to even if they do test for resistance.

Line 123- standard phrasing is “decreased sensitivity”

Dissemination as written could also be confusing for audiences outside of the vector biology/vector control space, as it means something different outside of these fields. Perhaps rephrase to “correlated with dengue susceptibility”.

Line 126- Would not bring up the metagenome here unless you discuss or reference the mosquito microbiota and insecticide resistance above. Otherwise, this is out of place.

Line 128- A couple of really important points are missing here. 1) chemical mosquito control is not species-specific. It kills everything, including things we want and need, like bees. Wolbachia and genetically engineered mosquito technologies are species specific and do not kill non-target organisms. This species-specificity is a huge benefit for this approach and incredibly so from a one-health standpoint. Chemicals indiscriminately kill lots of non-target organisms and potentially build up in the environment, accumulate in animals, etc.. Wolbachia and genetically engineered approaches (at least the ones tested in the field for mosquito control to date, such as sterile insect technique (SIT- genetically modified by radiation, or a new CRISPR-based SIT, such as the precision-guided SIT- not tested in the field for mosquito control (Aedes mosquitoes- Li et al 2021 Nature Communications; Li et al 2023 eLife) but it has been built in mosquitoes and trials have been done for agricultural pests or RIDL (Release of insects carrying a dominant lethal) do not. 2) Wolbachia (World Mosquito Program), SIT (IAEA) and RIDL (Oxitec) are all non-chemical technologies that have been tested in the field for Aedes aegypti/dengue control and should all be included here, since they have become parts of mosquito control efforts. Wolbachia is not genetically modified (see comments below), but the SIT and RIDL are genetically modified (SIT randomly modified by radiation, and RIDL engineered in the lab). 3) What other similar technologies are in development for mosquito control? Maybe mention technologies in the pipeline for field release and would have a one-health oriented benefit. For example, the precision guided-SIT is an SIT technology that uses CRISPR to sterilize male mosquitoes instead of radiation, so programs don’t need to use ionizing radiation to sterilize mosquitoes. These 3 points do not have to be discussed at length, but all should be mentioned and addressed.

Line 130- Wolbachia is not genetically modified. It does not change the mosquito genome. It is a naturally occurring bacteria that in the context of mosquito control are introduced to mosquito species to impact mosquito reproduction and possibly their ability to transmit pathogens.

Line 132- due to the emphasis on Aedes and discussion of some of its pathogens, I would not be so general here. Include the mosquito species and the arboviruses. These specifics are included elsewhere and are relevant here.

Line 144- drainage of standing water in marshes is environmentally devasting, so we essentially do not do it anymore. It also would have no impact on Aedes aegypti, which appears to be the primary focus on this article. Aedes aegypti larvae do not live in marsh environments.

Line 186- the AcTs model acronym should be spelled out for clarity (Citizen Action through Science)

Lines 186-188 Do you have any other examples? University Park, MD was where this study was conducted has a much higher-than-average income and education level. The beginning of this paragraph emphasizes “diverse sociopolitical circumstances”. This example is not that, or at least should be noted as contrary to this.

Lines 189 – 200- This would benefit from the inclusion of some mosquito control topic specific examples. For example, increasing engagement from the Spanish speaking community on genetically engineered mosquito technologies (Cheung et al. 2020 Global Public Health), but other examples exist.